# Does Zipf's law of abbreviation shape birdsong?

R. Tucker Gilman[1*], CD Durrant[2], Lucy Malpas[2,3], Rebecca N. Lewis[1,4]

**1** Department of Earth and Environmental Sciences, Faculty of Science and Engineering, The University of Manchester, Manchester, United Kingdom, **2** Faculty of Biology, Medicine, and Health, School of Biological Sciences, The University of Manchester, Manchester, United Kingdom, **3** Department of Natural Sciences, Manchester Metropolitan University, Manchester, United Kingdom, **4** Conservation and Science Team, Chester Zoo, Chester, United Kingdom

* tucker.gilman@manchester.ac.uk

## Abstract

In human languages, words that are used more frequently tend to be shorter than words that are used less frequently. This pattern is known as Zipf's law of abbreviation. It has been attributed to the principle of least effort – communication is more efficient when words that are used more frequently are easier to produce. Zipf's law of abbreviation appears to hold in all human languages, and recently attention has turned to whether it also holds in animal communication. In birdsong, which has been used as a model for human language learning and development, researchers have focused on whether more frequently used notes or phrases are shorter than those that are used less frequently. Because birdsong can be highly stereotyped, have high interindividual variation, and have phrase repertoires that are small relative to human language lexicons, studying Zipf's law of abbreviation in birdsong presents challenges that do not arise when studying human languages. In this paper, we describe a new method for assessing evidence for Zipf's law of abbreviation in birdsong, and we introduce the R package ZLAvian to implement this method. We used ZLAvian to study Zipf's law of abbreviation in the songs of 11 bird populations archived in the open-access repository Bird-DB. We did not find strong evidence for Zipf's law of abbreviation in any population when studied alone, but we found evidence for Zipf's law in a synthetic analysis across all populations. Overall, the negative concordance between phrase length and frequency of use in birdsong was several times weaker than the negative concordance between word length and frequency of use in written human languages. The method and the results we present here offer a new foundation for researchers studying if or how the principle of least effort shapes animal communication.

## Author summary

Since ancient times, people have been fascinated by birdsong, and imagined it to be the "language of birds." This analogy has become more exciting as

**Data availability statement:** All data and code used in this manuscript are available from https://doi.org/10.48420/24586791.v2.

**Funding:** RNL was partly funded by UK Natural Environment Research Council grant NE/L002469/1 "Training the Next Generation of Environmental Scientists." CDD was partly funded by the University of Manchester Merged Endowment Fund. The funders played no role in the study design, data collection, analysis, decision to publish, or preparation of the manuscript.

**Competing interests:** The authors have declared that no competing interests exist.

researchers have discovered that many genes and parts of the brain involved in birdsong learning and development are also involved in human speech. But, there is still much we do not know about how birdsong and human language are similar or different. Recently, researchers have been interested in whether Zipf's Law of Abbreviation (ZLA) holds in birdsong. In human language, ZLA says that words that are used more frequently tend to be shorter, because communication is more efficient if we have short words for the ideas we use most often. In birdsong, researchers have asked whether more frequently used notes are shorter, but results so far have been inconclusive. We developed a new computational tool for studying ZLA in birdsong and applied it to songs from 11 bird populations. We found evidence for ZLA in the set of populations we studied, but the pattern is weaker than in written human languages. More bird populations will need to be studied to confirm our results, and our computational tool will help researchers do that work.

## Introduction

Over the past three decades, birdsong has gained currency as a tractable model for studying how language develops and is transmitted in humans [1–6]. This has been due in part to the discovery of biological similarities between birdsong and human speech, including analogies in learning patterns [1,4,6], brain mechanisms [7], and regulatory genetics [1,8]. Birdsong is also amenable to experimentation that might be impractical or unethical in humans [1]. These attributes have made birdsong particularly appealing as a model system for studying human speech pathologies [9–11]. The growing importance of birdsong as a model of human language necessitates a clear understanding of how birdsong and human language are similar and how they differ, so we can better understand both the potential applications and the limitations of the model [12].

Zipf's law of abbreviation (ZLA) is a universal pattern in human languages [13]. ZLA states that words that are used more frequently tend to be shorter than words that are used less frequently [14]. This has been attributed to the principle of least effort. If an idea must be conveyed frequently, users will find or create shorter words to convey that idea, thus making communication more efficient [14–17]. If users must convey an idea only infrequently, then they can invest effort in longer words to ensure that the idea is communicated clearly [16]. Evidence supporting ZLA has been found in each of the nearly 1,000 human languages where it has been studied [13], and the law applies to both spoken language [18–22] and written characters [23,24]. Researchers have reported mixed support for ZLA in the vocal communication of other animals [25], including primates [19,26–30], cetaceans [31–33], bats [34], and hyraxes [35].

Relatively few studies have looked for patterns consistent with ZLA in birds. More than 30 years ago, Hailman and colleagues [36] reported that in black-capped chickadees (*Parus atricapillus*) shorter bouts of calls were more frequent than longer

bouts of calls, but they found no evidence that shorter call types were more frequent than longer call types. This has been cited as an example of ZLA in birds [17,27,29,37], but it is not clear that the pattern Hailman and colleagues [36] reported should emerge due to the mechanism Zipf [15] proposed. Birdsong or calls can be segmented into notes (continuous sounds separated by periods of silence), phrases (short series of notes that frequently or always appear together), calls or songs (series of notes or phrases separated by longer periods of silence), and bouts (series of often similar calls or songs separated by even longer periods of silence) [6,12,36]. If notes or phrases are analogous to words (which is debated [38]), then calls and bouts may be analogous to sentences and orations, respectively. ZLA posits a relationship between the frequency and length of words, but it is not clear that the same relationship should emerge at these higher levels. Indeed, a simple process in which birds begin bouts of calls and then decide independently after each call whether to stop or continue could produce patterns similar to those Hailman and colleagues reported [36]. In 2013, Ferrer-i-Cancho and Hernández-Fernández [19] found no evidence for ZLA in the calls of common ravens (*Corvus corax*) in data collected by Connor [39]. In 2020, Favaro and colleagues [40] reported that shorter note types appear more frequently than longer note types in the calls of captive African penguins (*Spheniscus demersus*). However, the study population used only three note types, so the perfect negative concordance between note duration and frequency of use that the authors observed could easily have arisen by chance. More recently, Lewis and colleagues [41] found no evidence for ZLA in the songs of a domesticated population of Java sparrows (*Padda oryzivora*), but Youngblood [42] reported evidence for ZLA in wild house finches (*Haemorhous mexicanus*). Thus, whether there are patterns consistent with ZLA in bird vocalizations remains an open question.

Given the mixed evidence for ZLA in bird vocalizations, one might reasonably ask whether we should expect to see ZLA in birdsong at all. In human languages, words have lexical meanings, and those meanings can be independent of the length of the word. For example, we can shorten "television" to "TV" or "telly" and the meaning does not change. In birdsong, the sound of a note may determine its value to the listener [12]. For example, in some species, females appear to interpret specific note types as indicators of male quality, perhaps because those note types are difficult to produce [43–46]. If a male produced shorter or longer versions of those note types, then the information conveyed to females about his quality might change. Thus, replacing long note types with short ones might not make communication more efficient but rather impede accurate communication among birds. This could prevent ZLA from emerging in birdsong. Thus, both empirically and theoretically, the questions of whether birdsong adheres to ZLA or should even be expected to adhere to ZLA are unanswered.

## Challenges to studying Zipf's law of abbreviation in birdsong

Assessing the evidence for ZLA in birdsong presents several challenges that we do not encounter when studying ZLA in humans. First, relative to the number of words in human languages, the number of note types used by most bird populations is small. A small number of note types makes it more difficult to detect a significant concordance between note type frequency and duration [13,19]. If the number of note types is very small, as in the calls of African penguins [40], then even a perfect concordance between the frequency and duration of note types may provide only weak evidence for ZLA. No amount of additional study can resolve this problem. Thus, we may never be able to say with confidence that ZLA exists in some populations. Instead, researchers interested in ZLA in birdsong may need to assess large numbers of populations and draw conclusions based on the full body of evidence.

A second challenge stems from the fact that different birds in the same population can have very different note type repertoires [47]. In humans, individuals in a population that shares a language are likely to use similar sets of words with similar frequencies. Thus, it may be reasonable to study ZLA at the population level. Researchers can select representative multiauthor texts and assess the concordance between the length and frequency of use of each word using simple rank correlations [13]. In contrast, in many bird species, individuals in the same population use different and sometimes non-overlapping sets of note types [47]. This makes it difficult to adequately sample the use of note

types in those populations. The problem is compounded by the fact that, in at least some species, song durations themselves appear to be constrained: birds that use longer note types sing fewer notes in each song [48,49]. In such species, if birds that sing shorter note types are at least as common as birds that sing longer note types, then we might see patterns consistent with ZLA at the population level even if no individual bird uses short note types more frequently than it uses long ones. However, such a pattern would not provide evidence for the principle of least effort proposed to underlie ZLA.

Because the principle of least effort suggests that individuals should use shorter types (i.e., words or notes) more frequently than longer ones, we might wish to look for ZLA at the level of individuals rather than populations. That is, if we choose a random individual from a population, are we likely to find that this individual uses shorter note types more frequently than longer ones? However, this question is made difficult by the fact that songs produced by individual birds in the same population may not be independent. In many species, songs are highly stereotyped and birds learn their songs from others [47,50,51]. If we find that two birds have note use consistent with ZLA, then the pattern may have arisen independently in each bird, or it may have arisen only once and both birds may have learned it from the same source. The second case is weaker evidence for ZLA. Thus, any attempt to study ZLA at the level of individuals must adequately account for the potential non-independence of individuals' songs.

Finally, perhaps the biggest challenge to studying ZLA in birdsong arises from the inherent difficulty of classifying notes. In human languages, especially in the written form, we can usually agree on whether two units represent the same word or different words [13,15,52]. In birdsong, determining whether two notes belong to the same note type is less straightforward. Notes are usually assigned to types by expert inspection of spectrograms [47,53] or sometimes by computational clustering [42,54–56]. Both methods are highly repeatable [47,56]. However, high repeatability does not ensure that the assigned note types match the intent of the birds that produced those notes. Different birds may produce notes that are very similar but are nonetheless objectively distinguishable among individuals (e.g., because they have slightly different peak frequencies or durations [47]). Should we assign these notes to the same or different types? Similarly, individual birds may produce objectively distinguishable versions of similar notes at different points in their song. In general, we cannot know whether the bird intends to produce slightly different notes, or whether it is attempting to produce the same note each time but its performance is constrained by the position of the note in the song. It is not clear that we can resolve this problem empirically. We could ask whether listening birds can distinguish between notes, but the ability of other birds to distinguish between notes does not necessarily indicate the intent of the producer. By analogy, I might attempt to imitate a word pronounced by my colleague, but listeners may still be able to distinguish my attempt from theirs. In some study populations (e.g., [47]), we may know which birds learned their songs from which others, and we may be able to use analogies in song structure to infer that similar-sounding notes are attempts to produce the same note type. However, for most populations, we do not know which birds learned from which others, and inferring the intent behind individual notes may be fundamentally beyond our grasp. In cases where it is difficult to decide whether notes should be assigned to the same or different types, it is likely that the ambiguous notes will have similar characteristics, and therefore the decision to split or merge types may have little effect on the durations of the note types. However, the decision to split or merge types will necessarily affect the frequencies with which those types appear, and so may affect our inferences about ZLA.

An especially difficult problem arises if classification errors are not independent of note durations. For example, we might be more likely to split longer note types, or to merge shorter note types, simply because longer note types give us more opportunity to identify potential differences among notes. If this is true, then our classification system will overestimate the frequencies of short note types and underestimate the frequencies of long note types in the population repertoire. This would produce a pattern consistent with ZLA not because of how birds use note types but rather because of how we perceive the note types they use.

## Overview of the current study

With these challenges in mind, Lewis and colleagues [41] developed a novel method for assessing ZLA in bird populations. This method studies ZLA at the individual level and is appropriate for systems where the repertoires of individual birds may be different and where frequencies of use may be learned and thus non-independent among birds. Here, we introduce the R [57] package ZLAvian (https://CRAN.R-project.org/package=ZLAvian) to implement Lewis and colleagues' [41] method. We apply ZLAvian to assess evidence for ZLA in 11 populations from 7 species of songbirds with songs archived in Bird-DB, an open access repository of annotated birdsong [58]. We synthesise results from the 11 populations as a first pass at quantifying ZLA in birdsong. Our work offers both a computational tool and an empirical foundation for studying ZLA in bird communication.

## Methods

In this section, we describe Lewis and colleagues' [41] method for studying ZLA in birdsong, as implemented in the R package ZLAvian. We formalize the method in Box 1. Then, we describe how we applied this analysis to study ZLA in the bird populations with songs archived in Bird-DB [58].

---

### Box 1. Formalisation of the method proposed by Lewis and colleagues

Let $n$ be the number of note types and $b$ be the number of birds in a dataset. Let $a_{ijk}$ be the log-transformed duration of instance $i$ of note type $j$ produced by bird $k$. Let $f_{jk}$ be the number of times bird $k$ produced note type $j$, and let $f_j$ be the total number of instances of note type $j$ across all birds in the dataset. Then, the mean log-transformed duration of note type $j$ as produced by bird $k$ is $\overline{a}_{jk} = \left(\frac{1}{f_{jk}}\right)\sum_{i=1}^{f_{jk}} a_{ijk}$. Let $\tau_k$ be the concordance (i.e., Kendall's $\tau_B$) between the $f_{jk}$s and the $\overline{a}_{jk}$s in bird $k$. Then, $\tau_k$ is a random variable with variance given by

$$v_k = \frac{2(2n_k + 5)}{9n_k(n_k - 1)}$$

where $n_k$ is the number of note types in the repertoire of bird $k$ [59]. The inverse-variance weighted mean concordance in the population is $\overline{\tau} = \frac{\sum_{k=1}^{b} \tau_k v_k^{-1}}{\sum_{k=1}^{b} v_k^{-1}}$. This is our test statistic for ZLA in the population.

Each $a_{ijk}$ differs from the expected log-transformed duration for note type $j$ in the population due to variation within and among birds. We obtain $\overline{a}_j$, the expected log-transformed duration for note type $j$ in the population, by fitting the random effects model

$$\mathbf{a}_{\cdot j\cdot} = \overline{a}_j + \mathbf{B_j}\gamma_\mathbf{j} + \varepsilon_\mathbf{j}$$

for each note type using the package lme4 in R [60]. Here, the vector $\mathbf{a}_{\cdot j\cdot}$ holds the log-transformed durations of all instances of note type $j$ in the data, and $\mathbf{B_j}$ is an $f_j \times b$ indicator matrix where entry $\mathbf{B_j}[p, q] = 1$ if the $p^{\text{th}}$ instance of note type $j$ was produced by bird $q$ and $\mathbf{B_j}[p, q] = 0$ otherwise. The vector $\gamma_\mathbf{j}$ has $b$ entries drawn from $N\left(0, \sigma_{\gamma j}^2\right)$ and the vector $\varepsilon_j$ has $f_j$ entries drawn from $N\left(0, \sigma_{\varepsilon j}^2\right)$. Thus, $\sigma_{\gamma j}^2$ and $\sigma_{\varepsilon j}^2$ are the variances in the mean log-transformed durations of note type $j$ among and within birds, respectively. Let $\mathbf{D}$ be the $n \times b$ deviation matrix in which entry $\mathbf{D}[j, k] = \overline{a}_{jk} - \overline{a}_j$ describes the deviation of the mean log-transformed duration of note type $j$ as produced by bird $k$ from the expectation for the population.

We can obtain one value of $\overline{\tau}$ under the null by permuting the log-transformed note durations among the note types without changing the set of note types that each bird produces. To do this, let $\mathbf{X}$ be an $n \times b$ matrix with each column equal to $\mathbf{x}$, where $\mathbf{x}$ is some permutation of the $\overline{a}_j$s. Then, the matrix $\mathbf{M} = \mathbf{X} \pm \mathbf{D}$ represents a resampling of the

---

mean log-transformed note type durations under the assumption that the note type duration is independent of the frequency of use, and where each bird's note type durations differ from the expected value for the population by the same magnitude as in the original data. Let $t_k$ be the concordance between the $M[j, k]$s and the $f_{jk}$s in bird $k$. Then, $\overline{\tau}_p = \frac{\sum_{k=1}^{b} t_k v_k^{-1}}{\sum_{k=1}^{b} v_k^{-1}}$ is one possible value of $\overline{\tau}$ under the null. We obtain a null distribution for $\overline{\tau}$ by repeated permutations.

## Statistical approach

ZLAvian works by computing the concordances between note type duration and frequency of use in individual birds sampled from a population, computing the weighted mean concordance across all birds in the sample as a test statistic, and comparing that test statistic to a null distribution obtained by permuting durations among note types while maintaining the song structure within and among birds. The analysis requires birdsong data in which notes have been assigned to types, the duration of each note has been measured, and each note can be attributed to an individual bird. In its default mode, ZLAvian computes and studies log-transformed note durations rather than working with raw note durations. This is important for the computation of the null distribution. We explain the decision to study log-transformed note durations after we explain how the null distribution is computed.

To obtain a test statistic for ZLA in a sampled population, ZLAvian computes the mean log-transformed duration of each note type as produced by each bird in the sample, and counts the number of times that each bird produced each note type. Then, it computes the concordance (i.e., Kendall's $\tau_B$) between the mean log-transformed duration and the frequency of use of note types within birds. This results in one value of $\tau_B$ for each bird. ZLAvian uses these $\tau_B$s to compute $\overline{\tau}$, the weighted mean value of $\tau_B$ in the population. In the default mode, ZLAvian weights each value of $\tau_B$ by its inverse variance [59]. Because $\tau_B$ is a random variable, weighting by the inverse variance accounts for the fact that the $\tau_B$s in birds with larger note type repertoires provide more accurate estimates of the population mean [61]. However, ZLAvian also offers users the option of computing $\overline{\tau}$ with each bird weighted equally. In either case, the computed $\overline{\tau}$ serves as a test statistic but also has a useful biological interpretation. Kendall's $\tau_B$, and therefore $\overline{\tau}$, ranges from −1 to 1 and is linearly related to the probability of concordance among observations. That is, if we were to randomly select a bird from the study population and then randomly select two note types as produced by that bird, then $(\overline{\tau} + 1)/2$ is the probability that the longer note type would appear more frequently. This makes $\overline{\tau}$ an intuitive metric for comparing the observed strength of ZLA across populations.

Next, ZLAvian computes a null distribution for $\overline{\tau}$. To do this, it first computes the expected log-transformed duration for each note type in the population. It estimates each expected log-transformed duration as the intercept in an intercept-only random effects model of the observed log-transformed durations for that note type, with a random effect for each bird that produced the note type. This method accords more weight to birds that produced the note type more frequently, because we can more accurately estimate the mean note type duration in those birds. ZLAvian then permutes the expected log-transformed durations among the note types at the population level. Thus, if a note type is assigned a particular duration by permutation, it is assigned that same duration in all birds that produced that note type. This permutation results in a set of population mean log-transformed durations for note types that we might see under the null hypothesis that note type durations and frequencies of use are independent, but it maintains the observed distribution of note type frequencies within and among birds in the population. This accounts for the fact that birds may learn note types and frequencies of use from other birds.

In nature, individual birds may produce the same note types in slightly different ways, and therefore the mean duration of each note type as produced by each bird will differ from the mean duration of that note type in the population. As a result, the same note types may have different rank order durations in different birds. How each bird produces each note type may be learned from other birds, and durations may not be independent among the note types a bird produces. For

example, a bird that produces a longer version of one note type may be more likely to produce longer versions of some other note type. We want to account for differences in note type durations among birds under maximally conservative assumptions about how note type versions are learned. To do this, ZLAvian computes the deviation of each bird's mean log-transformed note type duration from the population mean for each note type that the bird produced. This results in a deviation matrix with one entry per note type per bird. With equal probability, ZLAvian either adds the deviation matrix to or subtracts the deviation matrix from the permuted population mean log-transformed durations. This allows the rank order of note type durations to differ among birds, but maintains the structure of deviations in note type durations within and among birds. The result is a matrix of permuted and adjusted note type durations that might have been produced by birds in the sample if note type durations and frequencies of use were independent.

Finally, ZLAvian computes $\bar{\tau}_p$, the analog of $\bar{\tau}$ in the permuted and adjusted data. The null distribution for $\bar{\tau}$ is the set of $\bar{\tau}_p$s computed for every possible permutation of the mean log-transformed note type durations with the positive or negative deviation structures added. In most cases, the set of possible permutations is too large to compute all possible values of $\bar{\tau}_p$. Therefore, ZLAvian estimates the null distribution from a randomly chosen subset of the possible permutations. The p-value for the hypothesis that ZLA manifests in the study population is the proportion of the null distribution in which $\bar{\tau}_p$ is equal to or smaller than $\bar{\tau}$. Lewis and colleagues [41] developed this method to study the concordance between note type durations and frequencies of use in birdsong, but the method is transferable to other taxa and to other measures of production effort (e.g., bandwidth, concavity, excursion [42]).

The log transformation of note durations is important when the observed variability among birds in note type durations is reassigned to the note types after the population mean durations have been permuted. In many bird populations, the variability among individuals in the durations of particular note types scales with the population mean durations of those note types [48] (see also S1 Appendix). This is what we would expect if, for example, there are more opportunities for errors in duration to accumulate in longer note types. Furthermore, the differences among the durations of shorter note types tend to be smaller than the differences among the durations of longer note types – that is, the distributions of population mean note type durations are often right-skewed. If we were to permute the population mean raw durations among the note types and then reassign the original raw deviances to the permuted note type durations, we would often assign large deviances from long note types to note types that received short durations by permutation. As a result, the rank orders of note types with longer durations in the observed data would vary more among individuals in the permuted data than they did in the observed data, and the rank orders of note types with shorter durations in the observed data would vary less among individuals in the permuted data than they did in the observed data. This would change the relative importance of long and short note types in generating concordances at the population level. Log transformation reduces or removes the relationship between the mean and the interindividual variability of note type durations, and eliminates the right skew in the distribution of note type durations. Thus, we can permute log-transformed durations among note types without biasing interindividual variability in the rank orders of note types towards note types with longer observed durations. Some authors have advocated studying median rather than mean durations to assess ZLA in animal communication [20], but log transformation makes the distributions of durations approximately symmetrical, so means and medians are similar. Nonetheless, ZLAvian offers users the option to study ZLA using medians rather than means, and in the supplementary information we show that the qualitative results are similar in the populations we studied (S1 Table). ZLAvian also offers users the option to study raw rather than log-transformed measures of production effort. We do not advocate this when using durations to study ZLA, but it may be appropriate for users studying measures of production effort for which interindividual variability does not scale with the mean.

The method proposed by Lewis and colleagues [41] assesses evidence for ZLA within individuals while accounting for non-independence among individuals due to song learning or stereotypy. However, it does not correct for flaws in the classification of notes into types. Such flaws result in errors in the data, and in general statistical methods cannot correct such errors. However, we can assess how different kinds of note type misclassifications will affect our inferences

(S2 Appendix). If we incorrectly merge note types (i.e., assign notes to the same type when they should belong to different types), then we will overestimate the variance of the null distribution and our inferences will be conservative. If we incorrectly split note types, then we will underestimate the variance of the null distribution and our inferences will be anticonservative. Attempts to assess ZLA in birdsong must be interpreted in light of these potential biases.

## Application of ZLAvian to birdsong archived in Bird-DB

We downloaded 660 annotations of birdsong representing seven bird species (California thrasher, *Toxostoma redivivum*; redthroat, *Pyrrholaemus brunneus*; black-headed grosbeak, *Pheucticus melanocephalus*; sage thrasher, *Oreoscoptes montanus*; Cassin's vireo, *Vireo cassinii*; western tanager, *Piranga ludoviciana*; gray shrikethrush, *Colluricincla harmonica*) from the open access repository Bird-DB [58] on 18 April 2022. Annotations on Bird-DB can include one or more songs, but all songs in the same annotation are from the same bird. The phrases in each annotation are classified to types, and the starting and ending times for each phrase are reported. We used the reported starting and ending times to compute the phrase durations. The recordings represented in Bird-DB were collected on different days and at different locations, and we assumed that each annotation represents a different bird. Most phrases in Bird-DB are monosyllabic, and thus correspond to individual notes. A small number of phrases consist of short sequences of notes that typically appear together. Such polysyllabic phrases are often analysed as single units, and we follow this convention in the body of this paper. Our results are qualitatively similar if we divide the polysyllabic phrases into individual notes and use notes rather than phrases as the primary unit of analysis (S3 Appendix).

We cleaned the annotations downloaded from Bird-DB prior to analysis. Phrase types in Bird-DB are identified by two- or three-letter strings. We excluded any phrase with a type identifier that includes non-alphabetic characters, or that comprises fewer than two or more than three characters. These are likely to be data entry errors, and we cannot confidently assign these phrases to types. We also excluded annotations that include only one repeated monosyllabic phrase type. These annotations may represent alarm calls, and alarm calls may adhere to different rules than other calls or songs. Assessing concordance within annotations requires at least two phrase types, so no information about concordance was lost by excluding annotations that consisted of only one phrase type.

In some cases, songs from the same species on Bird-DB were annotated under different classification systems. We cannot analyse these songs together, because we do not know which phrases in one classification system correspond to which phrases in the other. If we treat phrases from different classification systems as different when they are in fact the same, we will overestimate the number of phrases in the species' repertoire and underestimate the variances of null distributions, and the p-values we obtain when testing for ZLA in that species will be anticonservative (S2 Appendix). Therefore, when songs from the same species were annotated using different classification systems, we treated annotations classified by each system as different populations. Tests conducted on multiple populations from the same species cannot be regarded as independent, because populations may share phrases and song structures. Nonetheless, tests of different populations that produce similar results can provide corroborating evidence for or against the presence of ZLA in that species.

For each population represented in Bird-DB, we report the number of annotations studied, the total number of phrase types across all annotations, the mean number of phrases and phrase types that appear in each annotation, the mean Shannon diversity of phrase types in annotations, the concordance between phrase type duration and frequency of use at the population level, and the mean concordance between phrase type duration and frequency of use by individuals in the population (i.e., $\overline{\tau}$). For population-level and individual-level concordances, we report the one-tailed p-values testing whether each concordance is more negative than we would expect under the null hypothesis that the note type duration and the frequency of use are unrelated. Thus, small p-values indicate patterns strongly consistent with ZLA and large p-values (i.e., close to 1) indicate patterns strongly contrary to ZLA. Finally, for each population we report the threshold

value of $\bar{\tau}$ that would be necessary to infer statistically significant ($\alpha = 0.05$) support for ZLA at the individual level. We call this the detection threshold. All reported measures were computed by ZLAvian.

Because phrase type repertoires in birdsong can be small, biologically plausible negative concordances between phrase durations and frequencies of use may not be statistically significant in individual bird populations, and researchers may need to synthesise across many populations to understand ZLA in birdsong. To do this, we fit an intercept-only model of $\bar{\tau}$ in the populations we studied, with a random effect of species to account for correlation among the concordances observed in populations of the same species. This analysis asks whether we would expect the concordance between phrase duration and frequency of use in a randomly selected bird species to be negative, even if concordances are not significantly negative when bird populations are studied individually.

If birdsong adheres to ZLA, we would like to know whether the strength of ZLA in birdsong is different from that in human languages. Following [13], we measured the length in characters and the frequency of use of words in 462 translations of the Universal Declaration of Human Rights downloaded from https://unicode.org/udhr/index.html on 21 May 2023. We computed the concordance between the length and the frequency of use of words in each translation. We compared the mean concordances we observed in our seven bird species to those we found in written human languages using a t-test with a Welch correction for unequal variance. This approach discards information because it averages $\bar{\tau}$ among populations within each bird species. Therefore, we also fit a Bayesian hierarchical model of $\bar{\tau}$ in birdsong and written human language, with a random effect of species in the birdsong data, and allowing unequal variance in the birdsong and human language data. We computed the probability of direction for the difference in the mean $\bar{\tau}$ in birdsong and written human language to confirm the inference from our t-test [62].

To understand how large phrase or note type repertoires in birdsong need to be for researchers to detect statistically significant evidence for ZLA in a population, we regressed the repertoire sizes in the populations we studied on the detection thresholds we achieved for each population, allowing random intercepts for each species. If researchers know the strength of ZLA they hope to detect in a population, this analysis allows them to estimate how large a phrase repertoire they would need for that strength of ZLA to be statistically significant.

## Results

We assessed the evidence for ZLA in 11 populations from 7 bird species (Table 1). Each population was represented by 2–296 annotations (mean 51.0, median 13, sd 87.2). The number of phrase types per population ranged from 9 to 748 (mean 188, median 114, sd 219) and the number of phrase types per annotation in the populations ranged from 2.8 to 89.5 (mean 30.0, median 24.9, sd 23.8).

At the population-level, the concordance between phrase type duration and frequency of use was negative (i.e., consistent with ZLA) in 8 of the 11 populations we studied (Table 1 and Fig 1). In two of these, western tanagers and one population of California thrashers, the p-values were less than 0.05 (i.e., strongly consistent with ZLA). Among the 3 positive population-level concordances, one in black-headed grosbeaks and one in Cassin's vireos had p-values greater than 0.95 (i.e., strongly contrary to ZLA). At the individual level, 10 of 11 concordances were negative, but only one (in western tanagers) had a p-value less than 0.05.

Across all populations, the expected mean individual concordance between phrase duration and frequency of use in birdsong was significantly negative ($\bar{\tau}$ = -0.071±0.031, p=0.028). The mean concordance we observed in human languages was -0.212±0.002. Thus, concordances were more negative in written human languages than in birdsong (p=0.001, probability of direction=0.001; Fig 2).

As expected, the repertoire size required to infer statistically significant ZLA in a population is larger when ZLA in the population is weaker (p<0.001; Fig 3). If birdsong in a population exhibited the expected strength of ZLA that we observed in our study ($\bar{\tau}$ = -0.071), we would expect to need a repertoire of approximately 194 phrases (95% confidence interval for the expectation, 143–260 phrases) to infer statistical significance.

**Table 1. Summary statistics and concordances in the songs of 11 populations of 7 bird species archived on Bird-DB.**

| Species | Records (birds) studied | Total phrase types | Phrases per record | Phrase types per record | Shannon diversity | Concordance (population) | Mean concordance (individual) | Maximum significant concordance |
|---|---|---|---|---|---|---|---|---|
| California thrasher | 89 | 748 | 145.7 | 14.4 | 2.20 | **−0.079 (p < 0.001)** | 0.022 (p = 0.820) | −0.039 |
| | 7 | 181 | 411.6 | 57.4 | 3.62 | −0.073 (p = 0.079) | −0.063 (p = 0.095) | −0.079 |
| Redthroat | 7 | 56 | 175.6 | 14.9 | 2.15 | −0.054 (p = 0.285) | −0.035 (p = 0.347) | −0.146 |
| Black-headed grosbeak | 83 | 451 | 153.4 | 27.5 | 2.92 | −0.009 (p = 0.388) | −0.046 (p = 0.064) | −0.049 |
| | 16 | 107 | 109.4 | 25.9 | 2.93 | **0.147 (p = 0.985)** | −0.036 (p = 0.298) | −0.109 |
| Sage thrasher | 2 | 147 | 234.0 | 89.5 | 4.13 | −0.049 (p = 0.209) | −0.040 (p = 0.259) | −0.102 |
| Cassin's vireo | 13 | 68 | 87.7 | 26.5 | 2.98 | **0.161 (p = 0.970)** | −0.063 (p = 0.211) | −0.127 |
| | 296 | 134 | 119.1 | 21.5 | 2.54 | −0.026 (p = 0.326) | −0.028 (p = 0.197) | −0.054 |
| | 41 | 114 | 94.6 | 26.6 | 2.85 | 0.083 (p = 0.903) | −0.032 (p = 0.219) | −0.069 |
| Western tanager | 3 | 56 | 128.7 | 22.7 | 2.30 | **−0.193 (p = 0.023)** | **−0.170 (p = 0.038)** | −0.157 |
| Grey shrike-thrush | 4 | 9 | 13.3 | 2.8 | 0.76 | −0.032 (p = 0.455) | −0.166 (p = 0.311) | −0.550 |

P-values less than 0.05 indicate patterns strongly consistent with ZLA (bold red), and p-values greater than 0.95 indicate patterns strongly contrary to ZLA (bold blue).

## Discussion

Across the bird species we studied, the expected concordance between phrase type duration and frequency of use was significantly negative, consistent with Zipf's law of abbreviation [14,15]. Within the individual populations, only 1 of 11 concordances was significantly negative, and this was without correcting for multiple testing. Thus, considering the populations individually, we cannot be confident that any particular population exhibits ZLA. Nonetheless, in 10 of 11 populations, the best estimates for the mean individual concordances were negative, and similar nonsignificant trends have been reported in the songs of Java sparrows [41] and in call repertoires at the population level in common ravens [19] and African penguins [40]. Taken together, this evidence is consistent with weak ZLA in bird vocalisations that is difficult to detect when phrase or note repertoires are small. It may be necessary to assess ZLA in many different bird species before we can draw clear conclusions about its existence or strength in birdsong generally. The results and tools we present here are an important step towards this goal.

Some of the populations we studied were represented by very small numbers of birds. When sample sizes are small, we cannot be confident that patterns we detect in the samples are general to the populations as a whole. For example, in western tanagers, we found significant evidence for ZLA in a sample of only three birds. We are confident that the repertoires of these birds conform to ZLA, but we cannot be confident that the repertoires of all birds in the population do. Nonetheless, the fact that we found negative concordances in samples from many different populations suggests that the pattern is unlikely to be due to stochasticity in sampling. Larger samples from more populations will be needed to confirm this result.

The birdsong phrases we analysed in our study were assigned to types by humans. We cannot know whether humans perceive phrases in the same way that birds do. When birds produce longer phrases, there may be more opportunities for

 

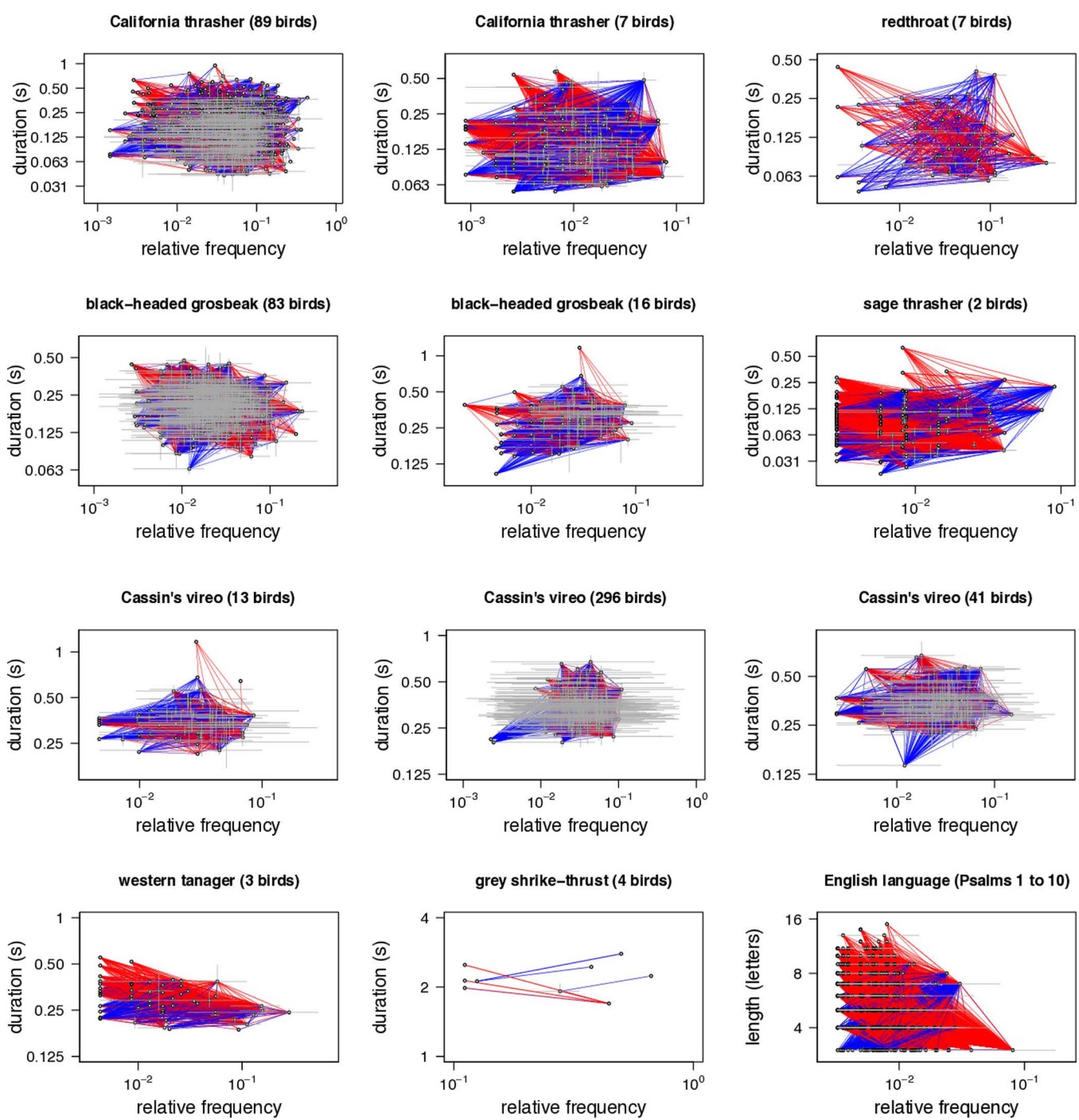

**Fig 1. Concordances between phrase type durations and frequencies of use in 11 bird populations.** Each open circle represents the mean frequency of use and the mean duration of one phrase type used in the population. Horizontal (vertical) grey lines show the range of relative frequencies of use (durations) among birds that used each phrase type. Phrase types that appear together in the repertoire of at least one bird are connected by colored lines. Heavier lines indicate that the pair of phrases was used by more birds. Blue (red) lines indicate that the concordance between frequency of use and duration was positive (negative) for that phrase pair. Intermediate colors indicate that the concordance was positive in some birds and negative in others. For example, this can occur when some birds use the phrase type more frequently than other birds. ZLA is present when the concordance between frequency of use and duration is negative. Thus, we should expect to see more red lines in figures for populations that adhere to ZLA. For comparison, the last panel shows the concordance between word length and frequency of use in English based on the first 10 psalms (American Standard Version), with each psalm treated as if it were produced by a different author. There is strong evidence for ZLA in English based on this sample ($\bar{\tau}$ = -0.253, p < 0.001).

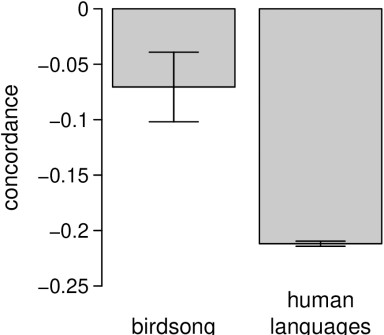

**Fig 2. Expected mean individual concordance between phrase type durations and frequencies of use in the songs of seven bird species, and concordance between word lengths and frequencies of use in written samples from 462 human languages.** Error bars show standard errors. Concordances are more negative in human languages than in birdsong (p = 0.001).

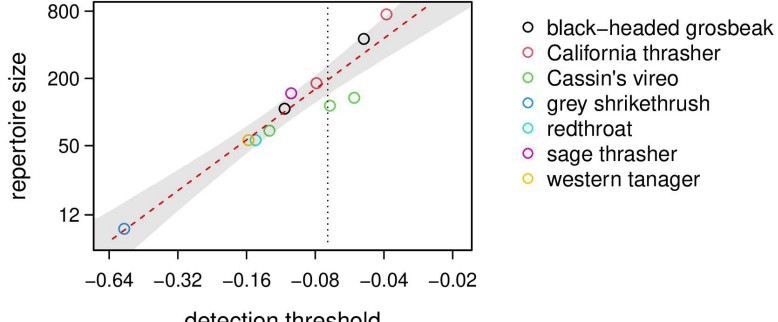

**Fig 3. The repertoire required to provide statistically significant evidence of ZLA in birdsong, plotted as a function of the strength of ZLA in the population.** Open circles show the detection thresholds and repertoire sizes for the populations we studied. The black dotted line shows the expected concordance across all populations. The red dashed line shows the predicted repertoire size needed to infer statistically significant evidence of ZLA in a birdsong sample with a given concordance between phrase duration and frequency of use. The shaded grey area shows the 95% confidence interval for the predicted repertoire size.

error than when they produce shorter phrases. If a bird attempts to produce the same phrase type several times, if some of those attempts include errors, and if researchers interpret phrases with and without errors as different types, then we may systematically overestimate the number of long phrase types and underestimate the frequency with which each long phrase type is used in that bird's repertoire. If researchers are less able to distinguish differences among phrase types when those phrase types are short, and if this leads researchers to merge phrase types that are different for birds, then we may systematically underestimate the number of short phrase types and overestimate the frequency of each short phrase type in birds' repertoires. In either case, our tests for ZLA will be anticonservative. Thus, misclassification of phrase types by humans could contribute to the negative concordances between phrase duration and frequency of use that we observed in bird populations.

We assessed the concordances between phrase duration and frequency of use in populations and also within individuals. Concordances at the population level sometimes differed qualitatively from those at the individual level. Patterns at the population level might arise if birds have different repertoires, and if some repertoires are more common or if birds with some repertoires are recorded more frequently than others. If individuals develop shorter versions of phrases they use more frequently, as proposed by Zipf [14,15], then we should expect concordances between phrase duration and

frequency of use within individuals. Many researchers have studied ZLA in animals at the population level [19,25–31,34,35,40,42], but our results underscore the importance of verifying these patterns at the individual level. Grzybek and Stadlober [63] have made a similar argument for the study of some laws that describe human languages.

The negative concordances between phrase duration and frequency of use that we observed in birdsong were several times weaker than those we observed between word length and frequency of use in written human languages (mean $\overline{\tau} = -0.071$ in birdsong vs mean $\overline{\tau} = -0.212$ in written human languages). Token length (i.e., phrase duration in birdsong or the number of characters in written words) was measured differently in the two systems, but we do not believe that differences in the how token lengths were measured explain the greater strength of ZLA in human languages. In studies of human languages, ZLA has been as strong or stronger when word length is measured by spoken duration than when it is measured in letters [21,22]. On the other hand, if token length is a less accurate measure of effort in birdsong than in written human language, then selection for a negative concordance between token length and frequency of use in birdsong might be weaker. We know of no study that has attempted to quantify the effort associated with token length in different communication systems. Alternatively or additionally, negative concordances might be weaker in birdsong than in human language because the function of the tokens differs in the two systems. In human languages, words have lexical meanings. By developing shorter versions of words they use more frequently, users of human language can communicate more efficiently [15]. In birdsong, notes or phrases may not have meanings independent of the notes or phrases themselves [12]. In the context of courtship or territory defence, the primary function of birdsong may be to advertise the quality of the singer, and notes or phrases that are more difficult to produce may indicate higher quality [46,64,65]. If this is true, then it may be impossible to shorten the duration of note types without also changing the message conveyed to listeners. This would disable the mechanism thought to promote ZLA in human languages. In some animals where patterns consistent with ZLA have been identified, the tokens studied are thought to have semantic meanings [26,66], making these communication systems more similar than birdsong to human language.

We used mean individual concordance (i.e., $\overline{\tau}$) as a test statistic in part because it is an intuitive measure of the strength of ZLA in a population. The p-values associated with $\overline{\tau}$ for each population measure the strength of evidence, but p-values by themselves are not evidence of strength. Even a weak negative concordance can be strongly significant if the population repertoire is large, and even a strong concordance can be nonsignificant if the population repertoire is small (Fig 3). Researchers who have studied ZLA in animals or in human languages have sometimes not reported strengths of concordance, and this may inhibit comparisons among systems and synthesis across studies. We urge authors of future work to report $\overline{\tau}$, or simply $\tau$ if they study ZLA at the population level, along with the p-values that evaluate its significance.

Detecting ZLA is difficult when note or phrase repertoires are small, and researchers may wonder how large a repertoire is needed to provide statistically significant evidence for a plausible strength of ZLA in a bird population. Fig 3 provides a first pass at answering this question. The precise repertoire size needed to provide significant evidence for ZLA in a population will also depend on the number of birds studied, the number of note or phrase types used by each bird, and how those notes or phrases are shared in the population, and will therefore be species- or even population-specific. However, inferences about ZLA in birdsong in general can be made using the strengths of concordance (i.e., the $\overline{\tau}$s) measured in populations, even if those $\overline{\tau}$s are not individually statistically significant. Therefore, we encourage researchers to compute and report $\overline{\tau}$ for bird populations they study even when population repertoire sizes are small.

In birds, alarm calls may offer an appealing context for studying ZLA. In some bird species, the phrases that make up alarm calls appear to have lexical meanings. For example, different phrases can indicate different predator types [67–69]. The phrases used in alarm calls can differ among populations and change within populations over time [70]. If birds can create shorter versions of alarm calls for predators they encounter frequently, then ZLA in alarm calls may become strong. Thus, studies of the concordance between phrase duration and frequency of use in alarm calls in bird populations under different predation pressures may reward effort. This work will require data on the durations of multiple phrase types used in alarm calls, and on the frequency with which those phrase types are used in multiple

populations. Obtaining frequencies of use for phrase types at the population level will require random sampling of population repertoires across all conditions that those populations encounter (e.g., soundscape recordings), and not just focal sampling under favorable recording conditions as in Bird-DB. We know of no system for which appropriate and sufficient data currently exists.

Identifying patterns consistent with ZLA in birdsong, and quantifying those patterns if they exist, may require studying the songs of many different bird populations and species. This requires songs that can be attributed to individual animals, where notes or phrases have been classified to type, and where the durations of notes have been measured. Annotated birdsong data already exists for many species, and automated note or phrase classification (e.g., [54,55]) may make such data easier to collect in the future. The ZLAvian package we introduce here will allow researchers who collect or maintain these data sets to test quickly and easily for evidence of ZLA. In this way, our work offers the opportunity to expand our understanding of ZLA and of the similarities and differences between birdsong and human language.

## Supporting information

**S1 Table. Evidence for ZLA in 11 populations of 7 bird species with songs archived on Bird-DB, computed with durations represented by medians rather than means.**
(PDF)

**S1 Appendix. Text, figure, and table illustrating how the standard deviation of phrase type durations scales with the mean.**
(PDF)

**S2 Appendix. Robustness analysis showing how the inferred relationship between phrase type duration and frequency of use depends on plausible types of phrase classification errors.**
(PDF)

**S3 Appendix. Robustness analysis showing qualitatively similar results when we use phrases as catalogued on Bird-DB or individual notes as tokens when assessing ZLA.**
(PDF)

## Acknowledgments

The authors thank Patrycja Strycharczuk for advice and discussion.

## Author contributions

**Conceptualization:** R. Tucker Gilman, Rebecca N. Lewis.

**Data curation:** R. Tucker Gilman, Lucy Malpas.

**Formal analysis:** R. Tucker Gilman, Lucy Malpas.

**Investigation:** R. Tucker Gilman.

**Methodology:** R. Tucker Gilman, CD Durrant.

**Project administration:** R. Tucker Gilman.

**Software:** R. Tucker Gilman, CD Durrant.

**Supervision:** R. Tucker Gilman.

**Validation:** R. Tucker Gilman.

**Visualization:** R. Tucker Gilman.

**Writing – original draft:** R. Tucker Gilman.

**Writing – review & editing:** R. Tucker Gilman, CD Durrant, Rebecca N. Lewis.

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
