## [Decision Letter · Decision Letter 0]

Dear Gilman,

Thank you very much for submitting your manuscript "Does Zipf's law of abbreviation shape birdsong?" for consideration at PLOS Computational Biology.

As with all papers reviewed by the journal, your manuscript was reviewed by members of the editorial board and by several independent reviewers. In light of the reviews (below this email), we would like to invite the resubmission of a significantly-revised version that takes into account the reviewers' comments.

We cannot make any decision about publication until we have seen the revised manuscript and your response to the reviewers' comments. Your revised manuscript is also likely to be sent to reviewers for further evaluation.

Sincerely,

Andrea E. Martin, Ph.D.

Academic Editor

PLOS Computational Biology

Natalia Komarova

Section Editor

PLOS Computational Biology

Reviewer's Responses to Questions

**Comments to the Authors:**

Reviewer #1: This paper asks whether Zipf’s law of abbreviation (ZLA), which is ubiquitous across languages, is present in birdsong as well. ZLA is based on the information theoretic concept that to most optimally convey information, more frequently used vocalizations will be shorter. Because ZLA describes optimal behavior in conveying information, and both the information content and units of information (e.g. syllables vs phrases) in birdsong are not well understood, this is a challenging problem, and the literature varies in whether ZLA is observed or not. Additionally birds generally produce unique vocal repertoires, thus individual identity needs to be taken into account when computing ZLA. To solve this problem, the authors implement a previously reported method for assessing ZLA in birdsong as an R package and test it on birdsong in the bird-db repository. They generally do not find any evidence for ZLA in the datasets, making the main analyses a null result. In the context of the current literature on ZLA in songbirds, this paper advances the field by implementing a common R library based on prior methods, but the actual analyses do not provide much information about whether ZLA exists in birds or not. The general conclusion is that this is hard to assess, but the authors never directly address why it is hard to do in birdsong - this seems like an approachable task given the methods they developed for the paper - for example looking at how many birds, how many notes/phrases, and how many unique notes/phrases are needed in a synthetic repertoire to observe LDA and comparing their empirical birdsong datasets to those measurements.

Specific points

I found the way that the authors motivate the expectation for ZLA in birdsong confusing and at times circular. ZLA proposes that common vocalizations will be shorter to maximize information. In birdsong, information contained in a song is not well understood. In this study, non-song vocalizations are removed from analysis, so the data is purely composed of birdsong. The authors motivate this analysis by saying that when a male is singing to a female, the information that is contained is fitness - longer and more complex vocalizations signal fitness, which is well established. But if the information conveyed in a song is tied to how long a vocalization is, then the expectation should be that ZLA does not exist in birdsong. But this is not conveyed.

The above point is related to the author's notion of “intent”, which is discussed but not defined. This seems fundamental to a question about the relationship between information-carrying units of birdsong and frequency. For example, if we were to segment speech into phonemes, syllables, or phrases, we presumably would not observe ZLA. The authors start to get at this issue by looking at different levels of organization for birdsong - many of the annotations in their dataset are composed of phrases rather than syllables, so they come up with a secondary way of doing the analyses by segmenting phrases into syllables. It seems the value of this analysis is in sweeping the range of possible ‘informative units’ in birdsong to see whether one conforms to ZLA or not. This second analysis is also mostly null, however, with similarly small effects in positive examples.

I see two major contributions of this work. The first is that the authors are taking a question that has been asked on a number of individual species and looking at a broad sampling of birds to see if a general trend emerges. The second is that the authors are presenting a statistical method and R package for looking for ZLA in symbolically segmented data like birdsong. For the large sampling approach - birddb, while a valuable resource, contains relatively small datasets sampled from the wild - would there be value in adding some more species, e.g. bringing in the data from the other species mentioned in the introduction? For the R package approach - for this to be a valuable resource the paper would need to additionally include a link to an online repository and online documentation (and might be better suited as a separate paper to something like JOSS).

A figure plotting the relationship between frequency and duration should be in the main text. For example, appendix 3, but for both notes and phrases.

Why is alpha set at 0.1 for significance?

The library/methodology the authors developed should be validated with synthetic data and language data beyond misclassification errors - for example, determining theoretical data requirements for observing ZLA in a dataset of birdsong.

On that note - how many unique note / phrase types are needed at minimum to do this analysis? Do all of the species meet these requirements?

Are we limited by the amount of data in these datasets? How did you decide when not to include an animal (i.e. when there is too little data for an animal). This is a recurring point in the text but is never addressed in a satisfying way.

The lack of figures and sectioning generally made it hard to follow the paper. It would aid the reader to see clear sections with supporting figures. For example, a structure where data is presented, models are defined, synthetic analyses are presented, and empirical data are analyzed. Currently the lack of supporting figures requires the reader to ‘take your word for it’ on a lot of the analysis.

Reviewer #2: This paper introduces a method to test whether birdsong confirms to Zipf’s Law of Abbreviation offering an accompanying R package. The paper is well written, and the methods and results are described clearly. It tackles a question that has been addressed by previous research although with somewhat conflicting results. Although the question is not novel and the research is not very original, there is definitely value in its approach, especially because it offers a way for other researchers to answer the question with their own data sets.

Not being a computational biologist myself, I am not qualified to comment on the appropriateness and details of the model. It seems straightforward enough to me, and as far as I can tell, there are no major issues with it.

I am going to comment on other aspects of the paper.

I have a basic issue with the study, which is that motivation for doing it in the first place is not well established. It struck me that the authors not only spent a long time discussing why it is difficult (even impossible in many cases) to classify and measure components in birdsong to test whether they conform to patterns predicted by ZLA, but they also detailed, pretty convincingly, the case against ZLA, why it probably wouldn’t make sense for it to evolve in birdsong. This is because longer and more complex songs have been shown to be appealing to females, and males who sing them often and loud will probably produce more offspring and will have more juveniles copying their songs (as male offspring tend to imitate their father’s or close relatives’ songs more than strangers’ songs). This would result in strong evolutionary pressures against short or simple songs and song syllables. With this in mind, the authors did not make a convincing case for doing the analysis, they say “Nonetheless, we believe such data is worth analysing” which does not seem like a good enough reason. I do agree that it’s interesting to look for commonalities between birdsong and human language, but there must be a strong conceptual motivation for it which has not been established in this paper.

Following on from the previous, I think it would have made more sense to exclude song syllables and focus only on calls which are used by both males and females and are much more dynamic and flexible in their use than songs. Moreover, songs are believed not to carry any sort of sophisticated meaning (besides “mate with me”) while calls probably convey more subtle social meaning, making them possibly more similar to words in human language. I don’t know if it’s possible to separate songs from social calls in the sample used in this study, but as it is, it seems to me that there isn’t a strong enough case for doing the analysis on the chosen sample.

I also wondered why the authors used songs that were annotated by hand when, as they acknowledged, sophisticated cluster analysis methods already exist for the classification of notes. Of course, these methods are not fool-proof either and subjective decisions must still be made about what features constitute a note or phrase, but the annotations used in this study seemed to contain variability which forced the authors to have to split up songs within a species into separate populations. Some of the species contributed only 2, 3 or 4 records, this seems too little for the analysis attempted by the authors. There was huge variation in the availability of data between the different species and populations, and this problem was not sufficiently addressed in the ms.

It is not clear why the authors used an alpha level of 0.1, and the interpretation of p values is very unusual. What was the null hypotheses for the tests? Not enough information is given on the statistical tests.

I was also a bit confused by the treatment of alarm calls. The authors excluded repeating notes from their analysis because, as they say, these may represent alarm calls that probably adhere to different rules than songs or calls. However, in the discussion, they make the point that alarm calls may make appealing targets for ZLA analysis. This contradicts the earlier decisions, and I wasn’t sure how to interpret it.

I appreciate that the authors were open and honest with the limitations of the study and the general approach, but the motivation, the sample and the analysis should have been better explained in the ms. All of these areas feel minimally addressed in the current draft.

Reviewer #3: This is very timely research as linguistic laws remain underexplored in birds (leaving aside penguins and ancient work by Hailman that needs to be revised as the authors correctly argue). In addition, bird song is crucial step as it deviates from previous research in the sense that it points to phonological syntax rather than to lexical syntax in Marler's sense.

To the best of my, knowledge, this is the widest exploration of Zipf's law of abbreviation in bird song that has ever been performed!

The article report weak support for the law of abbreviation (for most species, the sign of the correlation is negative as predicted by the principle of compression; however, the correlation is non-significant in most case). The article has various strengths: the wide range of species considered and the concerns about the impact of categorization on the emergence of the laws. However, I think that the clarity, methods and accuracy of arguments can be improved. I would love to see a substantially rewritten version. This article has the potential to become a milestone article for research in bird song and it is likely to boost research on linguistic laws within the bird song community.

p. 7-9 introduce "A method for assessing Zipf’s law of abbreviation in birdsong" but not in the methods section but in the introduction. Thus authors try to present critical methods following the literary style of of introductions. These pages are hard to understand because the authors have minimized the use of mathematical notation and replaced complex formulae by verbal explanations. Thse explanations are ambiguous and critical detail is missing. This method must be clearly explained in the methods section (with formulae and mathematical notation to guide explanations with precision). In the introduction, the authors may retain the essential information about the method. With the current vagueness of these pages, the results are difficult to interpret in depth.

That methodology is presented in a paper by Lewis et al that the authors cite. However, a reader of the present article should not be forced to read Lewis et al's article (ref 35) first. Lewis et al's article should be a back-up but not a must read. Having said that, I am totally sympathetic with the authors because I understand their challenge (I have faced that challenge in the past). I am sure they can fix it.

Abstract. "Zipf’s law of abbreviation predicts that in human languages, words that are used more frequently will be shorter than words that are used less frequently."

Zipf's law of abbreviation is an inductive pattern and thus it does not predict anything in a strict sense. It is important to distinguish between principles and their manifestations in the abstract and other parts of the article. The compression principle predicts Zipf's law of abbreviation. Zipf's law of abbreviation is one of the manifestations of the principle.

p. 2 "Zipf’s law of abbreviation (ZLA) states that, in human languages, words that are used more frequently tend to be shorter than words that are used less frequently." a reference to Zipf's (1949) classic book is required.

p.5. "Birds that use longer note types sing fewer notes in each song. In such species, if birds that sing shorter note types are at least as common as birds that sing longer note types, then we might see patterns consistent with ZLA at the population level even if no individual bird uses short note types more frequently than it uses long ones. However, such a pattern would not provide evidence for

the principle of least effort proposed to underlie ZLA."

"Birds that use longer note types sing fewer notes in each song" This seems an example of Menzerath's law (or Arens' law), which can be interpreted as manifestations of the principle of least effort (see Ref 30 and Gustison et al 2016 for the theoretical arguments). Lewis and collaborators have already investigated that law.

[METHODS]

p. 7 "We compute the mean logged duration of each note type as produced by each bird in the sample, and we count the number of times that each bird produced each note type."

The median is considered to give a more robust summary of the durations. See section 4.1 of Petrini et al 2023 for a justification based both on research on humans and other species.

The authors should justify the use of the mean and check if their conclusions would change and be more in line with the prediction of compression if means where replaced by medians.

The motivation of using logged duration instead of raw duration is unclear (logged duration). The Kendall tau correlation score will give the same value regardless of whether durations have been logged or not.

p. 7. "We compute \bar{\tau}, the population mean value of \tau with each bird weighted by the inverse variance of its \tau." Precise equation needed. I do not understand the meaning of inverse variance of \tau.

p. 8. "(\bar{\tau} + 1)/2 is the probability that the longer note type would appear more frequently." I am familiar with the mathematics of rank correlation but I cannot follow the argument. Kendall tau correlation is the probability that a pair of points in the sample are concordant. The authors are stating that half the probability that two pairs are concordant + 0.5 is the the probability that the longer note appears more frequently (for simplicity I have assumed Kendall tau a; Statistical packages such as R using Kendall tau b, which is a renormalized proportion).

I think that even a mathematically oriented reader needs help. Mathematical statements should be justified properly. Detailed arguments can be put in an appendix. Be careful with solutions that are just point to another paper where the lack of a proper mathematical argument remains.

p. 8. Second paragraph. I do not understand the null model. \tau should have an expected value of zero. Tau has an expected value of zero when one of the columns of the matrix is shuffled at random.

p. 9. I do not understand. Verbal explanations are imprecise.

One should not have to read the article by Lewis (ref. 35) to be able to follow the current article.

p. 10 and other places. The authors confront bird song against research in humans in ref [15] on written language. It would be useful to cite research on the law of abbreviation in spoken language, e.g., Petrini et al (2023) and across linguistic units, not only words (Hernandez-Fernandez et al 2019, Torre et al 2019).

[METHODS]

p. 13 the comparison of bird song (duration of vocalization) versus human language (word length) is weak in the sense the modality is different (vocal versus written) and the units of measurement (duration/time versus length in characters) is very different. There are two ways of addressing this: (a) using oral language and durations (see for instance Petrini et al 2023 or articles by Torre ) or (b) acknowledging the limitations of the control and nuancing the conclusions across the article. As reviewer, I am fine with just an accurate application of (b). Carrying out (a) controlling for speaker variation (the identity of the speaker is unknown in certain dataset; certainly not in Petrini et al 2013) and covering a sufficiently wide range of languages can be exceedingly complex.

In addition, I have not seen that the authors have controlled by number of tokens (the correlate of raw number of phrases in their analysis). The questions is if Zipf's law in humans becomes as weak as in birds when the same sample size (number of tokens) is the same as in birds. It is customary in quantitative linguistics to check if the findings still hold when using samples of same size for a fairer comparison (a subtle point is that it can be argued that using using only samples matched by number of tokens is not enough due to the scaling properties of symbolic sequences, e.g. a natural language text of 100 words may not have the same statistical properties of a text of 10000 words).

p. 15 Sign of the correlations. "However, in 10 of 11 populations, the best estimates for the mean individual concordance were negative." The extensions of information theory developed by Ferrer-i-Cancho and collaborators, Ref 30, predict that the correlation should be negative (non-positive) in case of optimal coding. Namely, the prediction was successful in 10 out of 11 species! There is a theory that justifies/motivates authors' quest for ZLA in birds.

p. 17. "The negative concordance between phrase duration and frequency of use that we observed in birdsong is several times weaker than the negative correlation between word length and frequency of use that we measured in human languages. This may indicate that birdsong and human language follow different organising principles, and suggest limitations in the value of birdsong as a model for human language learning or processing."

These statements need to be revised. As explained above, the comparison between human language and bird song has not been made using the same modality and the same units and have not been matched by token size. The current finds do not question of the same principles apply. The authors have found that the predictions of the principle of compression hold for 10 out 11 species! If the samples of human language were down-sampled to match the number of "tokens" of bird phrases, would the comparison of humans versus birds be that shocking?

Table 1. I do not understand the meaning of the columns "phrase record^-1" and "phrase types record^-1"? What is -1? It does not seem the inverse, i.e. x^{-1}=1/x. Apologize my ignorance. It may be some standard I am not aware of. Statistical tests are one-sided or two-sided? In the legend of Table S2.1 the test are said to be one-sided. Same issue in Table S2.1

Table 1 does not show the number of individual birds for each species. It is actually shocking that after rounding to leave one decimal, the column "phrases record", that is the average number of phrases, ends up showing integer numbers except in a few cases. I may be missing something.

Appendix 1. The authors present stochastic models that assume an exponential rank distribution for notes. Is this choice supported by the authors data or by previous research on bird song?

If not supported, the question is to what extend the conclusions of that Appendix depend on that assumption. That looks easy to check for the authors as they performed a note level analysis in Appendix 2. Consider also the work on duration of linguistic units other than "words" by Torre and colleagues, e.g. phonemes.

Appendix 2. p. 30 "and if ZLA operates in these populations it operates less strongly than in human languages."

ZLA is just a statistical pattern derived by induction from data. ZLA does not operate, compression does. Of course, one can argue that there alternative mechanisms to the action of the principle compression (minimization of mean durations).

Appendix 3. Fig. S3.1 shows plots in double logarithmic scale but the logarithmic scale dos not look a very professional one. It seems that the authors have take logs on x and y and then supplied log(x) and log(y) to the plotting tools. A professional plot in double logarithmic scale is obtained by supplying x and y to the plot function but then asking the scale to be logarithmic on both axes. In the authors plots, it is very difficult to know the true value of x and y (the authors do not indicate the base of the logarithm hence is log(x)=-4, the x could be 10^{-4}, e^{-4},...).

REFERENCES

Gustison ML, Semple S, Ferrer-I-Cancho R, Bergman TJ. Gelada vocal sequences follow Menzerath's linguistic law. Proc Natl Acad Sci U S A. 2016 May 10;113(19):E2750-8. doi: 10.1073/pnas.1522072113.

Hernández-Fernández, A., G. Torre, I., Garrido, J.-M., Lacasa, L. (2019). Linguistic laws in speech: The case of Catalan and Spanish. Entropy, 21(12). https://doi.org/10.3390/e21121153

Petrini, S.; Casas-i-Muñoz, A.; Cluet-i-Martinell, J.; Wang, M.; Bentz, C.; Ferrer-i-Cancho, R. Direct and indirect evidence of compression of word lengths. Zip's law of abbreviation revisited. Glottometrics, vol. 54, pp. 58-87, 2023.

Torre, I. G., Luque, B., Lacasa, L., Kello, C. T., Hernández-Fernández, A. (2019). On the physical origin of linguistic laws and lognormality in speech. Royal Society Open Science, 6(8), 191023. https://doi.org/10.1098/rsos.191023

**Have the authors made all data and (if applicable) computational code underlying the findings in their manuscript fully available?**

Reviewer #1: **No: ** Supporting data are available online, but I saw no link to a code repository or analyses

Reviewer #2: Yes

Reviewer #3: Yes

PLOS authors have the option to publish the peer review history of their article (what does this mean? ). If published, this will include your full peer review and any attached files.

**Do you want your identity to be public for this peer review?** For information about this choice, including consent withdrawal, please see our Privacy Policy .

Reviewer #1: No

Reviewer #2: No

Reviewer #3: No
---

## [Decision Letter · Decision Letter 1]

PCOMPBIOL-D-23-02047R1

Does Zipf's law of abbreviation shape birdsong?

PLOS Computational Biology

Dear Dr. Gilman,

Thank you for submitting your manuscript to PLOS Computational Biology. After careful consideration, we feel that it has merit but does not fully meet PLOS Computational Biology's publication criteria as it currently stands. Therefore, we invite you to submit a revised version of the manuscript that addresses the points raised during the review process.

As you will see, one of the reviewers has brought up concerns with regards to the statistical analysis implemented and making sure that the data support (or partially support) the claim. Please take those into account in your revision.

Please submit your revised manuscript within 60 days May 11 2025 11:59PM. If you will need more time than this to complete your revisions, please reply to this message or contact the journal office at ploscompbiol@plos.org. Please include the following items when submitting your revised manuscript:

We look forward to receiving your revised manuscript.

Kind regards,

Natalia L. Komarova

Section Editor

PLOS Computational Biology

**Journal Requirements:**

1) We note that your supporting information files are duplicated on your submission. Please remove any unnecessary or old files from your revision, and make sure that only those relevant to the current version of the manuscript are included.

2) We have noticed that you have uploaded Supporting Information files, but you have not included a list of legends. Please add a full list of legends for your Supporting Information files after the references list.

**Reviewers' comments:**

Reviewer's Responses to Questions

Reviewer #1: Although the authors have clearly put appreciable effort into revising their manuscript in light of the issues put forward by myself and the other reviewers, some of the main issues that I brought up in the last review have not been resolved in any substantive way. I am particularly concerned about the way that the results remain to be reported. For example, in the abstract, they still say that “we found weak trends consistent with Zipf’s law of abbreviation in 10 of the 11 populations we studied“ presumably referring to their observation of non-significant concordances observed in these populations. In combination with setting alpha at 0.1 in the previous revision, I am concerned that not reporting statistics in a principled way is a trend in this work, which is particularly important given that the main contribution of this work is in developing a statistical test. With that in mind I still believe that this work is a valuable contribution, and I ask that in a future version of this paper the authors will be more careful about ensuring that the claims they make align with their results.

Specific points:

The new analyses the authors made in reporting the strength of concordance needed to observe SLA given a dataset size is an important improvement in the current revision and I appreciate the authors adding this in.

As the authors note in their reply to my first review, this computed ‘maximum significant concordance’ is dependent on 3 factors: individual repertoire size, population repertoire size, and how similar repertoires are among birds. It would be valuable to report these numbers for some representative samples. For example, assuming a species had a 10%, 50% or 100% overlap in repertoires, you could plot a heatmap of the concordance needed for a range of individual and population repertoire sizes. Or similarity, holding the concordance at human level, you could plot what the required repertoire sizes would need to be. The point being - you don’t need to exhaustively sample these variables to give the reader a good sense of what sort of dataset they would need to collect from their species to determine whether ZLA is present.

I can see documentation for your code on CRAN. All I see in the github repo is a readme and a csv file. Please upload your code and put a minimal script (or notebook) that reads that CSV, runs your analysis, and reports the results on github.

I would strongly suggest the authors do not claim that they observe “weak trends”, which here appears to mean the same thing as “trending towards significance”. See https://doi.org/10.1136/bmj.g2215 for a discussion as to why. If the authors of this work are not clear in what the output of their method means, how are the researchers who are using their tool supposed to understand the results on their own data? If the authors want to “give readers enough information to decide how much credence to put in our results“ by avoiding significance testing, they should be more principled in reporting the level of evidence e.g. by using established information theoretic methods (See Burnham, Anderson, and Huyvaert, 2011;

https://link.springer.com/article/10.1007/s00265-010-1029-6)

The author states

“it may be that the only way to assess ZLA in birdsong generally is to study many populations and ask if weak trends in the direction of ZLA are more common than we would expect by chance. We explain this in lines 120-130 of the introduction” but also that “many labs and researchers have datasets from a range of species that could be analysed using our method.

Aggregating and analysing that data is beyond the scope of this study, but we

believe that the ZLAvian package will make such analyses easy.”

It would be valuable, at least in the discussion, to spend some time trying to estimate what effort future researchers would need to make to perform an analysis powerful enough to properly test for ZLA in birdsong. For example, which species produce a large enough repertoire to study ZLA?

Are there corrections for multiple tests being performed across species? Here there are 11 groups tested, and 2 are significant. I want to make sure this result is performing corrections to correct for multiple tests.

Reviewer #2: Thank you for taking my and the other reviewers' recommendations on board. I think the motivation could still be written in a more convincing way, but it isn't such a big issue that it should further delay publication. My other concerns have been addressed in the revised ms which is much improved, and I'm happy to recommend acceptance. As a small addition, I recommend including the following paper (just published last week) which shows Zipfian distribution in humpback whale songs as a citation: Inbal Arnon et al. (2025). Whale song shows language-like statistical structure. Science 387,649-653. DOI:10.1126/science.adq7055.

**Have the authors made all data and (if applicable) computational code underlying the findings in their manuscript fully available?**

Reviewer #1: **No: ** The R package is provided, the github repo is currently empty, and I cannot find the code for reproducing their results on these datasets.

Reviewer #2: Yes

PLOS authors have the option to publish the peer review history of their article (what does this mean? ). If published, this will include your full peer review and any attached files.

**Do you want your identity to be public for this peer review?** For information about this choice, including consent withdrawal, please see our Privacy Policy .

Reviewer #1: No

Reviewer #2: No

**Figure resubmission:**
---

## [Decision Letter · Decision Letter 2]

Dear Gilman,

We are pleased to inform you that your manuscript 'Does Zipf's law of abbreviation shape birdsong?' has been provisionally accepted for publication in PLOS Computational Biology.

Best regards,

Tobias Bollenbach

Section Editor

PLOS Computational Biology

As suggested by the reviewer, please include the figures you generated for the review in the manuscript.

Reviewer's Responses to Questions

**Comments to the Authors:**

Reviewer #1: I appreciate the effort the authors put into responding to my comments this time and I think the paper is much improved and I look forward to seeing it published.

The only remaining suggestion I have (and it is just a suggestion) is that readers who are planning on using your method would benefit from the inclusion of the the figures you generated for the review in the actual manuscript (Figure R1, R2, R3). I suspect one of the main reasons people will read the paper will be to get a sense of whether their dataset is sufficient to look for ZLA. These figures help answer that question.

**Have the authors made all data and (if applicable) computational code underlying the findings in their manuscript fully available?**

Reviewer #1: Yes

PLOS authors have the option to publish the peer review history of their article (what does this mean? ). If published, this will include your full peer review and any attached files.

**Do you want your identity to be public for this peer review?** For information about this choice, including consent withdrawal, please see our Privacy Policy .

Reviewer #1: No

---

## [Editor Report · Acceptance letter]

PCOMPBIOL-D-23-02047R2

Does Zipf's law of abbreviation shape birdsong?

Dear Dr Gilman,

I am pleased to inform you that your manuscript has been formally accepted for publication in PLOS Computational Biology. Your manuscript is now with our production department and you will be notified of the publication date in due course.

With kind regards,

Judit Kozma
